# GV1001 Inhibits the Severity of the Ligature-Induced Periodontitis and the Vascular Lipid Deposition Associated with the Periodontitis in Mice

**DOI:** 10.3390/ijms241612566

**Published:** 2023-08-08

**Authors:** Sharon Y. Kim, Yun-Jeong Kim, Suyang Kim, Mersedeh Momeni, Alicia Lee, Alexandra Treanor, Sangjae Kim, Reuben H. Kim, No-Hee Park

**Affiliations:** 1The Shapiro Family Laboratory of Viral Oncology and Aging Research, UCLA School of Dentistry, 714 Tiverton Ave, Los Angeles, CA 90095, USA; 2Department of Periodontology, Seoul National University Gwanak Dental Hospital, School of Dentistry and Dental Research Institute, Seoul National University, Seoul 08826, Republic of Korea; 3Teloid Inc., 920 Westholme Avenue, Los Angeles, CA 90024, USA; 4UCLA Jonsson Comprehensive Cancer Center, 10833 Le Conte Ave, Los Angeles, CA 90095, USA; 5Department of Medicine, David Geffen School of Medicine at UCLA, 10833 Le Conte Ave, Los Angeles, CA 90095, USA

**Keywords:** GV1001, periodontitis, systemic and vascular inflammation, atherosclerosis

## Abstract

GV1001, a 16 amino acid peptide derived from the catalytic segment of human telomerase reverse transcriptase, was developed as an anti-cancer vaccine. Subsequently, it was found to exhibit anti-inflammatory and anti-Alzheimer’s disease properties. Periodontitis is a risk factor for a variety of systemic diseases, including atherosclerosis, a process in which chronic systemic and vascular inflammation results in the formation of plaques containing lipids, macrophages, foam cells, and tissue debris on the vascular intima. Thus, we investigated the effect of GV1001 on the severity of ligature-induced periodontitis, vascular inflammation, and arterial lipid deposition in mice. GV1001 notably reduced the severity of ligature-induced periodontitis by inhibiting gingival and systemic inflammation, alveolar bone loss, and vascular inflammation in wild-type mice. It also significantly lowered the amount of lipid deposition in the arterial wall in *ApoE*-deficient mice receiving ligature placement without changing the serum lipid profile. In vitro, we found that GV1001 inhibited the Receptor Activator of NF-κB ligand (RANKL)-induced osteoclast formation and tumor necrosis factor-α (TNF-α)-induced phenotypic changes in endothelial cells. In conclusion, our study suggests that GV1001 prevents the exacerbation of periodontitis and atherosclerosis associated with periodontitis partly by inhibiting local, systemic, and vascular inflammation and phenotypic changes of vascular endothelial cells.

## 1. Introduction

GV1001, a 16 amino acid peptide derived from the catalytic site of human telomerase reverse transcriptase (hTERT), was developed as an anti-cancer vaccine to boost immune responses of CD8 and CD4 T cells [1]. GV1001 is known to exhibit many other biological activities, including anti-inflammatory, anti-apoptotic, antioxidant, and anti-viral properties [2,3]. Due to its anti-inflammatory effect, GV1001 was reported to prevent renal ischemia and myocardial ischemia-reperfusion injury [4,5,6]. Ko and her colleagues also reported that GV1001 induced an anti-inflammatory effect by downregulating the expression of tumor necrosis factor-α (TNF-α) and interleukin-6 (IL-6) in the *Porphyromonas gingivalis* lipopolysaccharide (LPS)-stimulated dental pulp cells without significant cytotoxicity [7].

Periodontitis is a chronic, multifactorial inflammatory response of the periodontium, which is associated with the development of several systemic disorders such as rheumatoid arthritis, psoriasis, systemic sclerosis, Alzheimer’s disease, and cardiovascular disease (CVD) in addition to destroying alveolar bone and tooth loss [8,9,10]. In particular, the association between periodontitis and cardiovascular disease, representative non-communicable diseases (NCDs), has received much attention. Many epidemiological studies have shown that periodontitis increases the risk of atherosclerosis [8,11]. It has also been shown that periodontitis can directly or indirectly induce systemic inflammation through biologically plausible translocated circulating oral microbiota, and those periodontal pathogens might promote atherogenesis in various animal models [12,13].

We previously reported that ligature-induced periodontitis exacerbates atherosclerosis in *Apolipoprotein E* (*ApoE*)-deficient mice [14,15,16] and also found that hyperlipidemia is required for the development and progression of atherosclerosis in mice with ligature-induced periodontitis [15,16,17].

As GV1001 has been reported to demonstrate antioxidant and anti-inflammatory activities [3,18,19], we investigated the effect of GV1001 on the severity of ligature-induced periodontitis and vascular inflammation in wild-type (WT) mice and the degree of lipid deposition in *ApoE*-deficient mice, respectively. To understand the mechanistic effects of GV1001 on periodontitis and periodontitis-associated atherosclerosis, we also investigated the in vitro effects of GV1001 on the receptor activator of nuclear factor kappa-B ligand (RANKL)-induced osteoclastogenesis and the TNF-α-induced phenotypic changes of vascular endothelial cells, major initiation and progression factors for atherosclerosis [17,20].

## 2. Results

### 2.1. GV1001 Reduced the Severity of Periodontitis Induced by the Ligature Placement in WT Mice

To investigate the effect of GV1001 on the development of periodontitis, forty-two 6-week-old WT mice were divided into six groups, as described in the Materials and Methods Section (Figure 1).

Microscopic findings revealed that the ligature placement notably increased periodontal attachment loss measured from the cementoenamel junction (CEJ) to the base of the pocket compared to the loss of the control mice receiving PBS injection without the ligature. Administration of 0.2 mg/kg GV1001 did not change the level of periodontal attachment loss compared to the control (PBS injection only). However, at the higher dose (1.0 mg/kg), GV1001 significantly reduced attachment loss induced by the ligature (Figure 2).

The micro-computed tomography (μCT) scanning analysis showed significant alveolar bone loss induced by the ligature placement. The average distance from the palatal and buccal CEJ to the alveolar bone crest (ABC) of the second molar was measured on the transverse cut of the 3-dimensional model produced through reconstruction after μCT. The administration of GV1001 without the ligature did not alter alveolar bone level compared to the control, but the degree of the alveolar bone loss was moderately but significantly reduced in mice receiving both the ligature with 1.0 mg/kg GV1001 group compared to that of the ligature with PBS injection (Figure 3).

To investigate the underlying mechanisms of the preventive effect of GV1001 on ligature-induced alveolar bone loss, we quantified the osteoclasts in the alveolar bone across different groups. As shown in Figure 4, the GV1001 reduced the number of osteoclasts compared to the control mice. As expected, the ligature significantly increased the number of osteoclasts in the alveolar bone, which was notably reduced by the GV1001 administration.

The immunofluorescent staining data revealed that GV1001 administration reduced the staining intensity of the ligature-induced increase of pro-inflammatory cytokines (e.g., TNF-α and Interleukin-6 (IL-6)) in the gingival tissue. A low level of TNF-α was found in the gingival tissue of control mice administered with PBS only, which was further reduced by the administration of GV1001. The ligature significantly enhanced TNF-α levels in gingival tissues, and GV1001 negated such an increase (Figure 5). Similarly, the ligature also notably enhanced the intensity of p50 staining, and a high dose of GV1001 (1.0 mg/kg) completely abolished the increase (Appendix A). Appendix A shows the increase in IL-6 staining intensity of IL-6 in gingival tissue, which is notably reduced by a higher dose of GV1001. These data indicate that GV1001 inhibits local inflammation caused by the ligature, resulting in the reduction of gingivitis and periodontitis severity in mice receiving the ligature placement.

### 2.2. GV1001 Inhibited RANKL-Induced Osteoclastogenesis In Vitro

To investigate whether the decrease in osteoclast numbers in vivo by GV1001 was directly associated with the inhibition of osteoclast formation, we evaluated the effect of GV1001 on the osteoclastogenesis of murine primary bone marrow cells in vitro. The results showed that GV1001 treatment significantly inhibited the formation of multinucleated osteoclasts induced by RANKL (Figure 6). These results indicate that GV1001’s preventive effect on ligature-induced alveolar bone loss might be partly due to GV1001’s inhibition of osteoclastogenesis.

### 2.3. GV1001 Prevented the Systemic Inflammation Induced by the Ligature Placement

As periodontitis is a local and systemic inflammatory condition, we assessed the effect of GV1001 on systemic inflammation by examining the levels of pro-inflammatory cytokines in serum. As shown in Figure 7, the ligature significantly increased the serum levels of pro-inflammatory cytokines, such as TNF-α, IL-1β, IL-6, and macrophage colony stimulating factor (MCSF). However, these increases were completely obliterated by GV1001, indicating that GV1001 prevented both the local and systemic inflammation caused by ligature-induced periodontitis.

To further validate the anti-inflammatory effect of GV1001 on the development of systemic inflammation from the ligature, we examined the cellular localization of NF-κB p50 in the splenic cells. Since the spleen serves as a primary origin of pro-inflammatory cytokines during systemic inflammation, we examined the cellular translocation of nuclear factor kappa B (NF-κB) p50 in mice subjected to ligature placement and GV1001 administration, as well as in the control mice. Most p50 were stained in the cytoplasm of splenic cells in the control group and the group receiving GV1001 without the ligature. The spleen cells of mice receiving the ligature showed noticeable translocation of p50 into the nucleus from the cytoplasm, which was reversed by GV1001 (1.0 mg/kg) administration (Appendix A). Moreover, the expression of TNF-α was notably increased by the ligature, and GV1001 almost completely blocked the expression of TNF-α in the splenic cells (Appendix A).

### 2.4. GV1001 Prevented the Vascular Inflammation Induced by the Ligature

As atherosclerosis is known to be induced by vascular inflammation [21], we studied the effect of GV1001 on vascular inflammation in WT mice by determining the expression levels of pro-inflammatory cytokines, such as IL-1β, TNF-α, and IL-6. As shown in Figure 8, the ligature placement moderately increased expression levels of pro-inflammatory cytokines in the arterial tissue. Such an increase was completely negated by GV1001, indicating that GV1001 prevented both systemic and vascular inflammation caused by the ligature.

### 2.5. Preventive Effect of GV1001 on the Exacerbation of Alveolar Bone Loss and the Lipid Deposition in the Arterial Wall Caused by the Ligature Placement in ApoE-Deficient Mice

We previously reported that the ligature around the molars of *ApoE*-deficient mice, not that of WT mice, exacerbated the arterial wall lipid deposition and atherosclerosis [15,16,17]. Thus, we investigated the effect of GV1001 on ligature-induced alveolar bone loss and the exacerbation of lipid accumulation in the arterial wall in *ApoE*-deficient mice (Appendix A). Like the WT mice, the ligature-induced severe alveolar bone loss, a hallmark of periodontitis, was also examined in the *ApoE*-deficient mice when analyzed with μCT (Appendix A). The amount of lipid deposition in the arterial wall was analyzed with an en face analysis. The full length of the aorta-to-iliac bifurcation was opened along the ventral midline and dissected free of the animal under a stereomicroscope. The analysis demonstrated minor lipid deposition in the arterial wall of control mice fed with a high-fat diet (HFD) for 5 weeks. Administration of GV1001 (2.0 mg/kg) did not alter the lipid deposition in mice without ligature placement. The ligature placement induced notably higher lipid accumulation in the arterial wall compared to the control mice by almost two-fold, and the increase was completely blocked by 2.0 mg/kg GV1001 administration in *ApoE*-deficient mice (Figure 9).

### 2.6. GV1001 Did Not Alter the Cholesterol Profiles of Mice in Serum

As shown in Appendix A, the serum lipid profiles of mice receiving PBS, GV1001, ligature plus PBS, and ligature plus GV1001 were not different. These data indicate that the preventive effect of GV1001 on the increased ligature-induced lipid deposition in the arterial wall is not related to the altered cholesterol profile.

### 2.7. GV1001 Inhibited the TNF-α-Induced Phenotypic Changes of Endothelial Cells In Vitro

Numerous studies [22,23,24] have shown that atherosclerosis is initiated by the disruption of endothelial function through endothelial-mesenchymal transition (EndMT). Previous findings [15,17,25] also demonstrated that TNF-α, a major proinflammatory cytokine implicated in atherosclerosis development, induced the endothelial-mesenchymal transition (EndMT) in human umbilical vein endothelial cells (HUVEC), resulting in the impairment of endothelial barrier function. Since GV1001 significantly reduced systemic and vascular inflammation and lipid depositions within the arterial wall, we investigated the effect of GV1001 on TNF-α-induced EndMT in HUVECs, aiming to elucidate the underlying mechanism of GV1001’s anti-atherosclerotic effect. As shown in Figure 10, TNF-α-induced a notable increase in the number of cells stained with FSP-1, indicating a notable EndMT by TNF-α. GV1001 significantly and dose-dependently inhibited the TNF-α-induced number of HUVEC converted to mesenchymal cells. These findings demonstrate that GV1001 may inhibit the enhanced arterial lipid deposition and atherosclerosis caused by the ligature partly via the disruption of the endothelial cell barrier in arterial walls, thereby effectively inhibiting atherogenesis.

As shown in Figure 11, the transwell migration assay also confirmed the inhibitory effect of GV1001 on the TNF-α-induced EndMT of HUVEC dose-dependently. As expected, TNF-α increased the number of migrated HUVECs. The low dose (5 μg/mL) of GV1001 mildly inhibited the migration, and the high dose (10 μg/mL) of GV1001 in a culture medium significantly inhibited the TNF-α-induced migration of the cells which has mesenchymal cell phenotypes.

## 3. Discussion

Although GV1001 was initially developed as an anti-cancer therapeutic agent, increasing lines of evidence suggest that it also has anti-inflammatory functions. In this study, the direct role of GV1001 on inflammation-associated diseases, such as periodontitis, and atherosclerosis, was investigated by examining the severity of ligature-induced periodontitis, vascular inflammation, and arterial lipid deposition in mice. GV1001 notably reduced the severity of ligature-induced periodontitis by inhibiting gingival and systemic inflammation, alveolar bone loss, and vascular inflammation in WT and *ApoE*-deficient mice. In vitro, GV1001 inhibited the RANKL-induced osteoclast formation and TNF-α-induced EndMT in endothelial cells. Our study suggests that GV1001 is a putative therapeutic agent that can reduce disease burdens of inflammatory origins such as periodontitis and atherosclerosis.

To investigate the effect of GV1001 on ligature-induced periodontitis and atherosclerosis, we used both WT and *ApoE*-deficient mice. Since *ApoE*-deficient mice show hyperlipidemic and systemic proinflammatory properties, we used WT mice to minimize the effect of the genetic and systemic factors in investigating the effect of GV1001 on ligature-induced periodontitis. However, as hyperlipidemia is necessary to evaluate the degree of arterial lipid deposition, we used *ApoE*-deficient mice to assess the effect of GV1001 on arterial lipid accumulation/atherosclerosis.

Our study demonstrates that GV1001 inhibits the severity of ligature-induced periodontitis by inhibiting local and systemic inflammation and suppressing osteoclast formation in the alveolar bone. Although GV1001 significantly reduced local inflammation and osteoclastogenesis, we noted that GV1001 mildly reduced alveolar bone loss. The placement of ligatures around teeth might lead to continuous mechanical tissue irritation and destruction, and this physical factor would be a reason for the lack of prevention of bone loss with GV1001.

GV1001 also showed a significant systemic anti-inflammatory effect in mice receiving the ligature, indicating that GV1001 effectively counteracted local and systemic inflammation induced by the ligature. Thus, the inhibition of the arterial lipid deposition by GV1001 in the mice with the ligature might partly be due to the mitigation of systemic inflammation by GV1001.

Since the NF-κB activation is caused partly by the binding of TNF-α to TNF-receptors of cells [26,27,28], TNF-α derived from the sites of local inflammation by the ligature might travel to the bloodstream and bind to the TNF-receptors of splenic cells. As the translocated p50 transactivates target genes, including those encoding many pro-inflammatory cytokines [29], the systemic inflammation caused by the ligature could be amplified. In fact, we noted a high expression of TNF-α in the spleen. However, additional mechanistic studies investigating the relationship between periodontitis and the activation of NF-κB in the splenic cells will be necessary to understand the detailed mechanisms of systemic inflammation induced by periodontitis. Moreover, inhibition of NF-κB activation by GV1001 in the spleen needs further studies to understand the mechanisms of GV1001’s systemic anti-inflammatory activity. Although previous studies have shown the anti-inflammatory effects of GV1001 [3,4], our study is the first to demonstrate that GV1001 effectively reduces the severity of periodontitis caused by rigorous challenges, such as ligature placement.

Atherosclerosis is the process of plaque formation, including lipids, macrophages, foam cells, and tissue debris in the vascular intima, caused by chronic systemic and vascular inflammation. We previously reported that the ligature placement around molars induced an enhanced accumulation of lipids in the arterial wall via systemic and vascular inflammation [15,16]. From the present study, we noted that GV1001 notably inhibited lipid accumulation but did not alter the serum lipid profile when compared to the control mice or mice with the ligature, indicating that the anti-lipid deposition effect of GV1001 is not related to the alteration of cholesterol profiles in serum. Instead, it may be due to its anti-inflammation properties and other reasons. The endothelial-to-mesenchymal transition (EndMT) of aortic endothelial cells contributes to the initiation and progression of atherosclerosis by inducing the accumulation of lipids, monocytes, macrophages, and other leukocytes in the intima of the arterial wall, and we previously reported that the ligature placement around molars induced EndMT in vitro [17]. Thus, we investigated the effect of GV1001 on the TNF-α-induced EndMT using HUVEC cells and found that GV1001 notably inhibited the EndMT process induced by TNF-α, a major proinflammatory cytokine responsible for the development and progression of atherosclerosis. Furthermore, GV1001 demonstrates its anti-atherosclerotic effect by partly inhibiting systemic and vascular inflammation and anti-EndMT activities.

In a recent study using hyperlipidemic *ApoE*-deficient mice, infection with *Porphyromonas gingivalis* and multiple microbial experimental infections (*Porphyromonas gingivalis*, *Treponema denticola*, *Tannerella forsythia*, and *Fusobacterium nucleatum*) resulted in the enhanced oxidative stress response generated within aortic endothelial cells, induction of toll-like receptor (TLR) and inflammasome signaling [12]. Therefore, further research utilizing such a model is warranted to investigate and validate the effect of GV1001 on atherogenesis, thereby providing a better understanding of the underlying mechanisms of the disease.

## 4. Materials and Methods

### 4.1. Animals and Animal Welfare Considerations

A total of forty-two 6-week-old WT male mice (C57BL6 background) were purchased from the Jackson Laboratory (Bar Harbor, ME, USA). All mice were housed in a pathogen-free animal experimental facility at the University of California, Los Angeles University, under a 12 h light/dark cycle. All mice were fed normal chow and had free access to drinking water and food. The health and behavior of the mice were monitored three times a week throughout the whole duration of the experiment (6 weeks). A mixture of ketamine (100 mg/kg; VetOne, NADA #045-290, Boise, ID, USA) and xylazine (5 mg/kg; Akron, NADA #139-236, Lake Forest, IL, USA) were used as anesthetics during ligature placement and phosphate-buffered saline (PBS) or bacteria inoculation. Carprofen (3 mg/kg; Norbrook Labs, NDC 55529-131-01, Newry, UK), a pain relief drug, was also used after ligature placement to minimize the pain of the mice. The ketamine/xylazine mixture and carprofen were administered via intraperitoneal injection, and isoflurane was administered via inhalation. All mice were administered ketamine/xylazine prior to euthanasia to minimize suffering. Euthanasia was performed via cardiac perfusion, and the heartbeat of the mice was assessed for 5 min to verify death. All procedures were performed in compliance with the institution’s policy and applicable provisions of the United States Department of Agriculture (USDA) Animal Welfare Act Regulations and the Public Health Service (PHS) Policy. The Animal Research Committee (ARC) of the University of California, Los Angeles (UCLA) approved the experimental protocols under ARC# 2019-057. The mice were divided into six groups: (1) phosphate-buffered saline (PBS) s.c. injection (*n* = 5); (2) GV1001 (0.2 mg/kg) injection (*n* = 5); (3) GV1001 (1.0 mg/kg) injection (*n* = 5); (4) ligature placement around the maxillary second molars and PBS s.c. injection (*n* = 7); (5) ligature placement and GV1001 (0.2 mg/kg) injection (*n* = 10); and (6) ligature placement and GV1001 (1.0 mg/kg) injection (*n* = 10). PBS and GV1001 were subcutaneously injected 3 times per week for six weeks. On week 3, the ligature placement was performed by placing a silk suture around the 2nd maxillary molars (M2). The mice were sacrificed at the end of the experimental period, and the severity of periodontitis was determined with histological examinations, gingival inflammation, alveolar bone loss, and systemic inflammation (Figure 1). GV1001 (Tetromotide, A001/SF375) was obtained from GemVax/Kale Co., Ltd. (Seongnam-si, Gyeonggi-do, Republic of Korea).

### 4.2. Ligature Placement for the Creation of Periodontitis

Under the ketamine/xylazine (100 and 5 mg per kg, respectively) mixture and carprofen anesthesia, 6.0 silk sutures were placed subgingivally around the second molars for three weeks, as described previously [30]. Dental examinations were performed daily to ensure the ligatures were intact.

### 4.3. Apolipoprotein E-Deficient Mice Model

Nine-week-old *ApoE*-deficient male mice (C57BL/6J, Jackson Lab, Barr Harbor, ME, USA) were fed with a high-fat diet (HFD) (#D12079B, Research Diets, New Brunswick, NJ, USA) for one week. One week after starting the HFD, they were divided into four groups: (1) phosphate-buffered saline (PBS) injection (*n* = 5); (2) GV1001 (2.0 mg/kg) injection (*n* = 5); (3) PBS injection and silk ligature placement around the maxillary second molars (*n* = 5); and (4) ligature placement and GV1001 (2.0 mg/kg) injection (*n* = 5). The PBS or GV1001 was subcutaneously injected three times a week for the whole period. On week 2, a silk suture was placed around the maxillary second molars (M2) to perform ligation, and 4 weeks later, the mice were sacrificed. Ligature was placed under general anesthesia using ketamine/xylazine, as described previously (Figure 2). Unlike the studies evaluating GV1001’s effect on the severity of ligature-induced periodontitis in WT mice, we used *ApoE*-deficient mice for appraising the effect of GV1001 on arterial lipid deposition due to the necessity of hyperlipidemia for assessing the amount of lipid deposition as previously reported [16].

### 4.4. Tissue Collection

Whole blood was collected from mice by cardiac puncture under general anesthesia with isoflurane (VetOne. G46D22, Boise, ID, USA). The mice were then perfused and fixed with 4% paraformaldehyde in phosphate-buffered saline (PBS) via the left ventricle for 5 min. After perfusion, the entire length of the aorta-to-iliac bifurcation was exposed, carefully dissected from surrounding tissue and stored in RNAlater (Thermo Fisher Scientific, AM7020, Waltham, MA, USA). The spleen was also resected and fixed with 4% paraformaldehyde in PBS, pH 7.4. The maxillae were excised, and half of the palatal tissues were harvested using a blade. The harvested palatal tissues were stored in RNAlater at −80 °C and used to determine the expression of proinflammatory cytokines. For the μCT analysis, the maxilla bones were fixed with 4% paraformaldehyde in PBS, pH 7.4, at 4 °C overnight and stored in 70% ethanol solution.

### 4.5. μCT Analysis

The fixed maxillae were subjected to μCT scanning (Skyscan1275, Bruker-microCT, Kontich, Belgium) using a voxel size of 20 μm^3^ and a 1 mm aluminum filter at 60 kVp and 166 μA. Two-dimensional slices from each maxilla were finely tuned and combined using NRecon and CTVox programs (Bruker, Cambridge, UK) to form a three-dimensional reconstruction. The region of interest was specifically set by viewing the orthogonal projections of each slice through the Dataviewer program (Bruker, Cambridge, UK). The level of bone resorption was quantified by measuring the distance from the palatal and mid-buccal cement-enamel junction (CEJ) to the alveolar bone crest (ABC) in the transverse section of the second molars using the CTAn program (Bruker, Cambridge, UK). Measurements were performed by a periodontist (YK), and readings were checked blindly by another investigator (SYK).

### 4.6. Histological and Immunofluorescence Analysis

After μCT scanning, the maxillae were decalcified with 5% EDTA and 4% sucrose in PBS (pH 7.4) for three weeks at 4 °C. The decalcification solution was changed daily. Decalcified maxillae and sectioned aorta were sent to the UCLA Translational Procurement Core Laboratory (TPCL, Los Angeles, CA, USA) and processed for paraffin embedding. Blocks were sectioned at 5 μm intervals using a Microtome (Thermo Fisher Scientific, HM355S, Waltham, MA, USA), and slides were dewaxed in xylene to stain with Hematoxylin and Eosin (H&E). For tartrate-resistant acid phosphatase (TRAP) staining, the sections of maxillae were stained using a solution made with Napthol AS-TR phosphate sodium salt (Sigma-Aldrich, SLBS8237, St. Louis, MO, USA) and Fast Violet Red dye (Sigma-Aldrich, 32348-81-5, St. Louis, MO, USA) in buffer and then counterstained with hematoxylin. Liver and spleen tissues were also sectioned at 3 μm intervals after paraffin embedding, and H&E staining was performed. The full length of the aorta-to-iliac bifurcation was stained with Oil Red O (Sigma-Aldrich, 1320-06-5, St. Louis, MO, USA) as previously described [31] and pinned out flat, intimal side up, between cover slides. Aortic images were captured with a digital camera (Nikon-D7500 DSLR Camera, Tokyo, Japan), and the atherosclerotic lesion size was determined by ImageJ software version 1.48 (NIH, Bethesda, MD, USA). The digital images of the histochemically stained section were obtained using the microscope (Olympus, DP72, Tokyo, Japan) and also analyzed by ImageJ.

For immunofluorescent staining, the paraffin sections of maxillae were incubated with primary antibodies, TNF-α (Abcam, ab6671, Cambridge, UK), IL-6 (Thermo Fisher Scientific, 701028, Canoga Park, CA, USA), IL-1β (Thermo Fisher Scientific, P420B, Canoga Park, CA, USA), or NF-κB p50 (Thermo Fisher Scientific, 51-0500, Canoga Park, CA, USA), followed by fluorometric detection with Alexa Fluor 594-conjugated secondary antibodies (Thermo Fisher Scientific, A-11012, Canoga Park, CA, USA). Sequentially, the sections were mounted on slides with VECTASHIELD TM anti-fade mounting medium with 4′,6-diamidino-2-phenylindole (DAPI) (Vector Laboratories, H1200, Burlingame, CA, USA). IF pictures were taken using a confocal fluorescent microscope (Carl Zeiss, LSM 700, Oberkochen, Germany).

### 4.7. Osteoclastogenesis Assay

The effect of GV1001 on osteoclastogenesis was performed in murine primary bone marrow macrophage cells. Primary bone marrow cells were isolated from mice’s femur and tibia and grown in Dulbecco’s Modified Eagle medium (DMEM; Thermo Fisher Scientific, 11965092, Waltham, MA, USA) with 10% fetal bovine serum (FBS; Thermo Fisher Scientific, A5256701, Waltham, MA, USA). Osteoclastogenesis was induced by culturing the cells in the presence of mouse macrophage-colony stimulating factor (M-CSF; R&D Systems, 416-ML-050, Allendale, NJ, USA) for two days, followed by exposure to RANKL (R&D Systems, 462-TEC-010/CF, Allendale, NJ, USA) with or without GV1001 (0, 1, 5, or 10 μg/mL) for five days. Then, the cells were fixed with ice-cold 70% ethanol for 15 min and stained for tartrate-resistant acid phosphatase (TRAP) staining kit (Sigma-Aldrich, 387A, St. Louis, MO, USA), a marker of osteoclasts, for five minutes.

### 4.8. Serum Lipid and Cytokine Measurements

Levels of triglyceride, total cholesterol, high-density lipoprotein (HDL), low-density lipoprotein (LDL), and very low-density lipoprotein (VLDL) were measured using enzymatic assay kits in the UCLA Cardiovascular Core Facility [17]. The whole mice’s blood was collected with cardiac puncture. The serum was separated from the blood for the detection of pro-inflammatory cytokines using the Quantibody Mouse Cytokine Array kit (RayBiotech, Inc., QAM-CYT-1, Peachtree Corners, GA, USA), which allows the determination of low levels of TNF-α, IL-1β, IL-6, and MCSF from the serum samples.

### 4.9. Quantitative Real-Time Polymerase Chain Reaction

Total RNA from mouse tissues was extracted using RNAqueous total RNA isolation kit (Invitrogen, AM1914, Carlsbad, CA, USA) and reverse-transcribed using iScript™ Reverse Transcription Supermix (Applied Biosystems, A25741, San Francisco, CA, USA). Subsequently, real-time quantitative reverse transcription-polymerase chain reaction (qRT-PCR) was performed using PowerUp™ SYBR Green Master Mix (Thermo Fisher Scientific, A25742, Waltham, MA, USA) according to the manufacturer’s protocol. The sequences of the primers used for qRT-PCR are described in the Appendix A in the Online Appendix A. GAPDH served as control, and the fold induction was calculated using the comparative ΔCt method and presented as relative transcript levels (2^−ΔCt^).

### 4.10. En Face Analysis

The full length of the aorta-to-iliac bifurcation was opened along the ventral midline and dissected free of the animal under a stereomicroscope (Zeiss, Stemi 305, Oberkochen, Germany). For en face analysis, the aorta was stained with Sudan IV (Sigma-Aldrich, St. Louis, MO, USA) as previously described [32,33] and pinned out flat, intimal side up, between cover slides. Aortic images were captured with a Nikon digital camera (Nikon-D7500 DSLR Camera, Tokyo, Japan) and analyzed using ImageJ software version 1.48 (NIH, Bethesda, MD, USA; http://imagej.nih.gov/ij (accessed on 11 May 2023)).

### 4.11. Cell Culture and Reagents

Human umbilical vein endothelial cells (HUVEC; Lonza, C2519A, Basel, Switzerland) were cultured in endothelial basal medium-2 containing EGM-2 SingleQuot Kit (Lonza, CC-3162, Basel, Switzerland). The medium was renewed every 48 h. Cells were cultured at 37 °C and in a CO_2_ air atmosphere with a humidity of 5% (*v*/*v*).

### 4.12. Induction of EndMT

Thirty thousand human umbilical venous endothelial cells (HUVEC) were seeded in a chamber culture well of a 4-well chamber slide (Lab-Tek II, Nunc, C6932, Rochester, NY, USA). One day after the plating of the cells, the cells were exposed to 5 μg/mL or 10 μg/mL GV1001 in an endothelial growth medium for 3 h. Then the cells were cultured in the culture media containing 100 ng/mL TNF-α with or without GV1001 for 20 h. The cells were fixed with ice-cold 100% methanol (Fisher Scientific, A412P-4, Waltham, MA, USA) for 10 min and permeabilized with 0.1% PBS-Tween20 (Sigma-Aldrich, P2287, St. Louis, MO, USA) for 5 min. Then, the cells were blocked for 1 h with 10% normal goat serum at room temperature. Primary antibodies (5 μg/mL of anti-VE-Cadherin; Abcam, ab33168, and 10 μg/mL of anti-FSP-1, ab218512; Abcam) were applied overnight at 4 °C. Corresponding fluorescence-tagged secondary antibodies were applied for 1 h at room temperature (Alexa Fluor 594-conjugated secondary antibody (Thermo Fisher Scientific, A-11032, Canoga Park, CA, USA) and Alexa Fluor 488-conjugated secondary antibody (Thermo Fisher Scientific, A-11034, Canoga Park, CA, USA). The cells were washed with PBS three times for 5 min between each step. The cells were mounted using VECTASHIELD^TM^ anti-fade mounting medium with DAPI (Vector Laboratories, H1200,Burlingame, CA, USA). Immunostaining was observed under a Confocal microscope (Carl Zeiss, LSM 700, Oberkochen, Germany).

### 4.13. Cell Migration Assay

Ten thousand HUVEC cells were seeded in twenty-four transwell plates with 8 μm pore size membrane (Sigma-Aldrich, 3422, St. Louis, MO, USA). The cells were pre-treated with 5 μg/mL or 10 μg/mL GV1001 in endothelial basal medium containing 0.5% FBS for 4 h. The complete medium with 100 ng/mL TNF-α was placed in the lower well and was incubated overnight. The cells inside the insert were removed with a cotton swab, and the migrated cells on the bottom of the insert were stained with 0.5% crystal violet (Sigma-Aldrich, C0775, St. Louis, MO, USA) in PBS with methanol. The migrated cells were micro-photographed in 100× with the Olympus digital microscope (Olympus, DP72, Tokyo, Japan).

### 4.14. Statistical Analyses

All graphs were created using GraphPad Prism software, and statistical analyses were calculated using a GraphPad software (GraphPad Prism 9 Software, Boston, MA, USA). For multiple comparisons, one-way ANOVA with Newman–Keuls test was used. A *p*-value of less than 0.05 was considered significant. All results from in vitro were confirmed by at least three independent experiments. Error bars represent mean ± SEM.

## Figures and Tables

**Figure 1 ijms-24-12566-f001:**
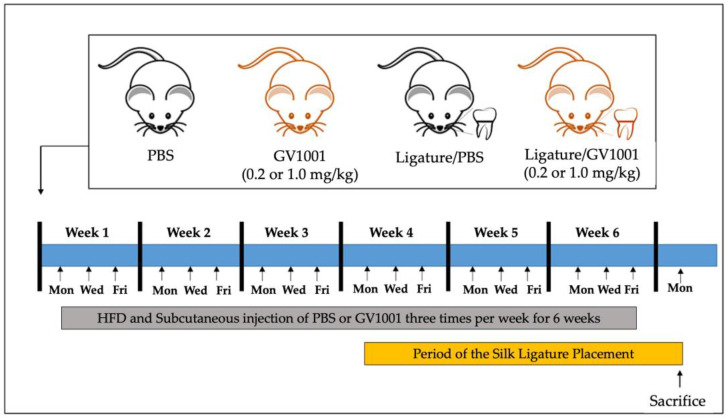
Experimental design for investigating the effect of GV1001 on the severity of ligature-induced periodontitis. Six-week-old mice were divided into six groups: (1) control mice receiving subcutaneous (s.c.) injection of phosphate-buffered saline (PBS) thrice per week for six weeks; (2) mice receiving GV1001 (0.2 mg/kg) s.c. injection thrice per week for six weeks; (3) mice receiving GV1001 (1.0 mg/kg) s.c. injection thrice per week for six weeks; (4) mice receiving PBS injection thrice per week for 6 weeks and the silk ligature placement around maxillary second molars three weeks after the initiation of PBS injection; (5) mice receiving GV1001 (0.2 mg/kg) s.c. injection thrice per week for six weeks and the ligature placement around maxillary second molars three weeks after the initiation of GV1001 injection; and (6) mice receiving GV1001 (1.0 mg/kg) s.c. injection three times per week for six weeks and the ligature placement around maxillary second molars three weeks after the initiation of GV1001 injection.

**Figure 2 ijms-24-12566-f002:**
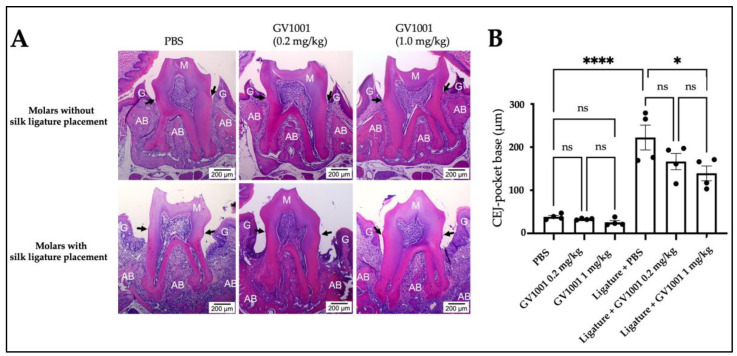
Representative Hematoxylin and Eosin (H&E) staining of the periodontium of maxillary second molar. (**A**) 1.0 mg/kg GV1001 administration notably reduced the periodontal attachment loss, while 0.2 mg/kg GV1001 administration did not alter it compared to the control (scale bar: 200 μm). The dark arrow indicates the CEJ. (**B**) The average periodontal attachment loss (the distance from the buccal and palatal CEJ to the pocket base) of the second molar. Results represent the means ± SEM performed in four samples. Statistical analysis was performed with one-way analysis of variance (ANOVA). * *p* < 0.05; **** *p* < 0.0001; ns = not significantly different between two groups (*p* > 0.05); CEJ and dark arrows, cement-enamel junction; M—molar tooth; G—gingiva; AB—alveolar bone. Black dots indicates individual animal samples.

**Figure 3 ijms-24-12566-f003:**
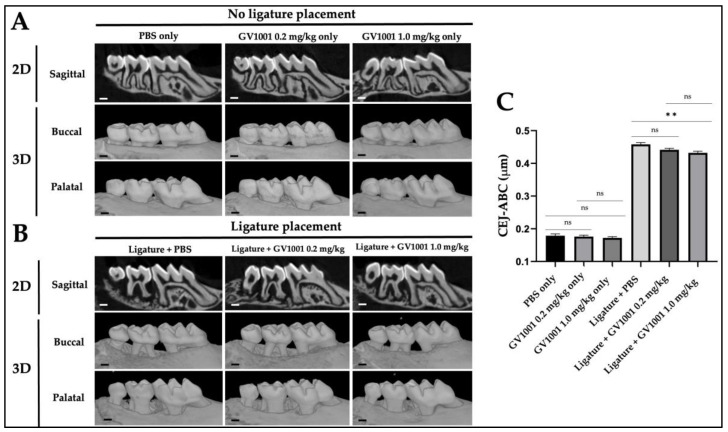
Representative two- or three-dimensional μCT images of mice maxillae (**A**) with no ligature placement and with/without GV1001 injection. (**B**) with the ligature placement and with/without GV1001 injection. Scale bar: 0.1 mm. (**C**) The average distance from the palatal and buccal CEJ to the ABC of the second molar. Results represent the means ± SEM performed in ten samples. Statistical analysis was performed with one-way ANOVA. ** *p* < 0.01; ns—not significantly different between two groups (*p* > 0.05); CEJ—cement-enamel junction; ABC—alveolar bone crest; 2D—two dimensional; 3D—three dimensional.

**Figure 4 ijms-24-12566-f004:**
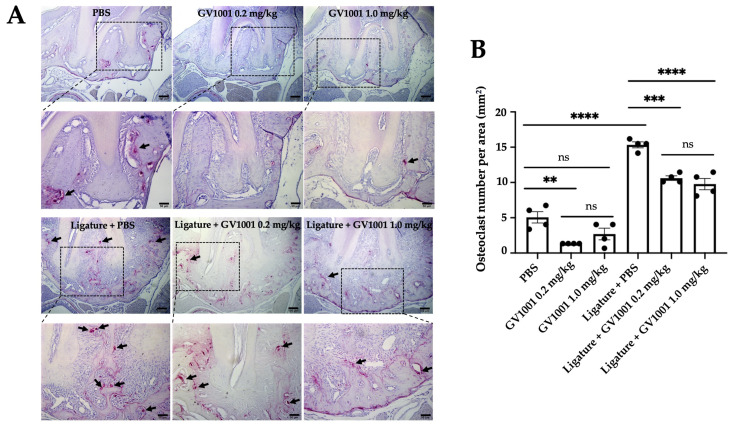
Tartrate-resistant acid phosphatase (TRAP) staining of periodontium of mice with or without ligature placement, along with PBS or GV1001 administration. (**A**) Dark arrows indicate osteoclasts. (**B**) The number of osteoclasts per surface area (mm^2^) and the results represent the means ± SEM performed in four samples. Statistical analysis was performed with one-way ANOVA. ns—not significantly different between two groups (*p* > 0.05); ** *p* < 0.01; *** *p* < 0.001; and **** *p* < 0.0001.

**Figure 5 ijms-24-12566-f005:**
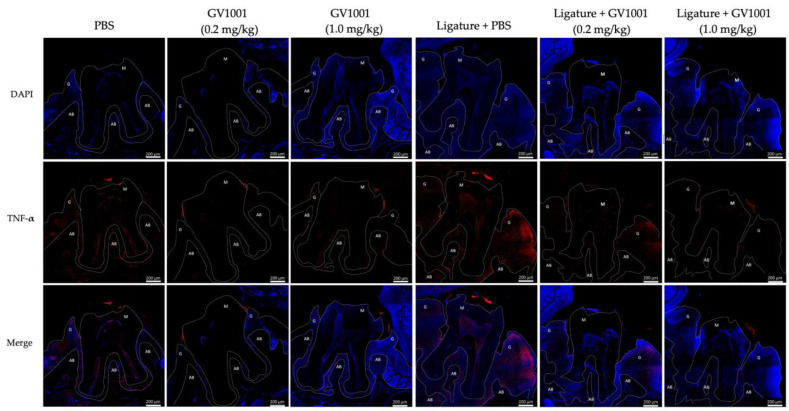
Representative immunofluorescent staining of TNF-α of gingival tissue. A low level of TNF-α was found in the gingival tissue of control mice receiving PBS only, and the protein level was notably reduced by GV1001 administration. The ligature highly enhanced the level of TNF-α in the gingival tissue (red color), and GV1001 administration completely negated the increase induced by the ligature placement. The blue dots are nuclei of cells, and the red areas are TNF-α. G: gingiva, M: molar, AB: alveolar bone.

**Figure 6 ijms-24-12566-f006:**
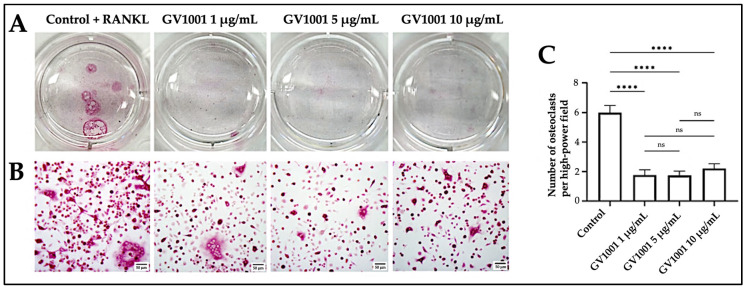
GV1001 inhibits the differentiation of murine primary bone marrow macrophages to osteoclasts. (**A**) Well images; (**B**) representative images of high-power fields of primary bone marrow cells treated with 0, 1, 5, or 10 μg/mL of GV1001; and (**C**) Quantitative analysis showing the number of osteoclasts per high-power field averaged from 3 wells per group. Statistical analysis was performed with one-way ANOVA. Data are shown as mean ± SEM. **** *p* < 0.0001, ns—not significantly different between two groups (*p* > 0.05). The scale bar represents 50 μm.

**Figure 7 ijms-24-12566-f007:**
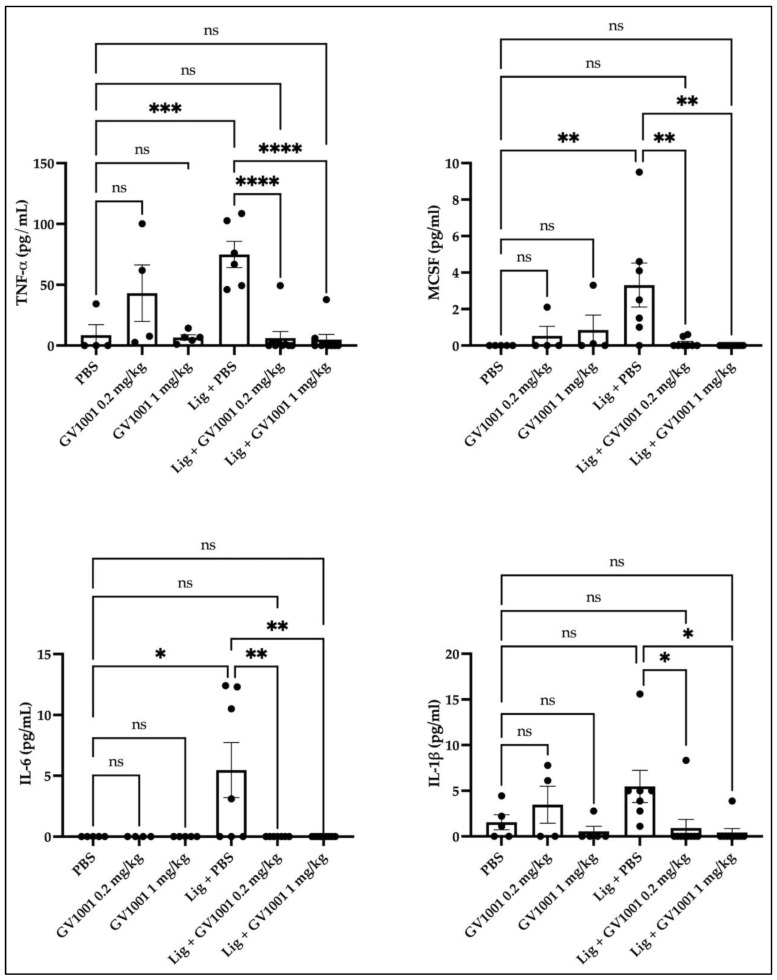
The ligature notably increased the serum levels of pro-inflammatory cytokines, and GV1001 completely inverted such an increase. Levels of TNF-α, IL-1β, and IL-6 from the mouse serum were measured using pre-coated enzyme-linked immunosorbent assay (ELISA) plates. Glyceraldehyde 3-phosphate dehydrogenase (GAPDH) served as a loading control. Statistical analysis was performed with one-way ANOVA. * *p* < 0.05, ** *p* < 0.01, *** *p* <0.001, and **** *p* < 0.0001. ns—not significantly different between two groups.

**Figure 8 ijms-24-12566-f008:**
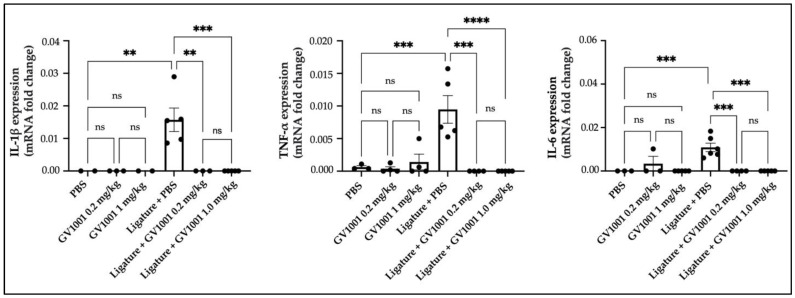
The ligature placement moderately increased the expression levels of IL-1β, TNF-α, and IL-6 pro-inflammatory cytokines in arterial tissue, and GV1001 completely blocked such increases. The gene expression was measured with qRT-PCR. GAPDH served as a loading control. Statistical analysis was performed with one-way ANOVA. ns—not significantly different between two groups (*p* > 0.05), ** *p* < 0.01, *** *p* < 0.001, and **** *p* < 0.0001. Results represent the means ± SEM performed in 2–5 samples.

**Figure 9 ijms-24-12566-f009:**
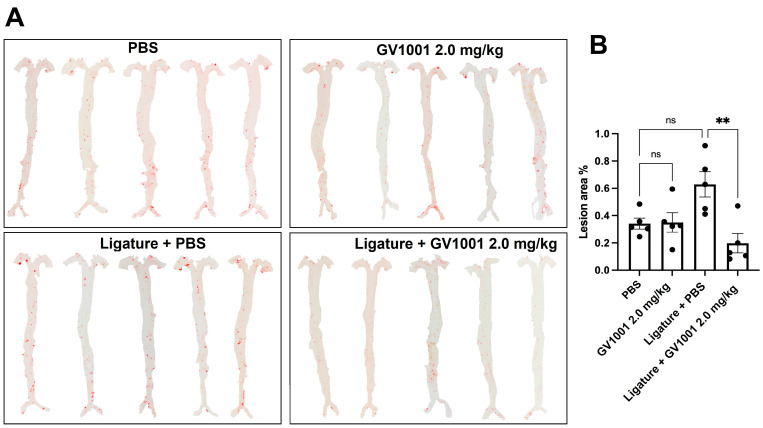
En face analysis of mice arteries. GV1001 administration did not alter the lipid deposition in mice without ligature placement. The ligature placement notably increased the lipid deposition on the arterial wall compared to the control (PBS-only group), and GV1001 (2.0 mg/kg) completely prevented the ligature-induced lipid deposition. (**A**) Representative photographs of mice arteries from the en face preparation after staining with Sudan IV (5 mice per group). (**B**) Quantification of areas stained by Sudan IV. ** *p* < 0.01, ns—not significantly different between two groups (*p* > 0.05), results represent the means ± SEM performed in five samples.

**Figure 10 ijms-24-12566-f010:**
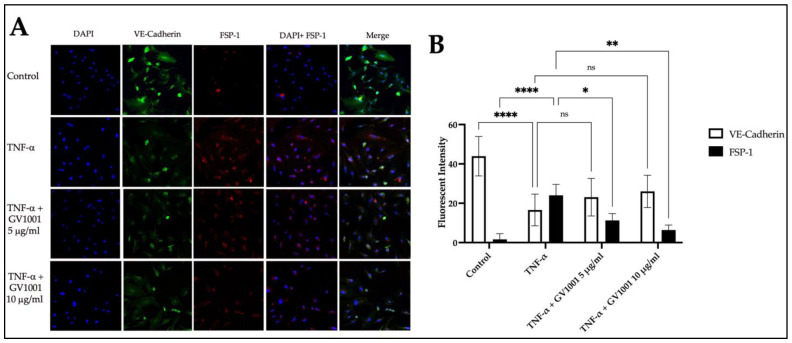
(**A**) Representative immunofluorescence staining of HUVEC with anti-VE-Cadherin antibody (Green: endothelial cell marker) and anti-FSP-1 (Red: mesenchymal cell marker) antibodies. Nuclei were counterstained with DAPI (Blue). (**B**) Quantification of areas stained by VE-Cadherin or FSP-1. **** *p* < 0.0001, ** *p* < 0.01, and * *p* < 0.05, ns—not significantly different between two groups (*p* > 0.05). Results represent the means ± SEM performed in four samples. The magnifications of the figures were ×200.

**Figure 11 ijms-24-12566-f011:**
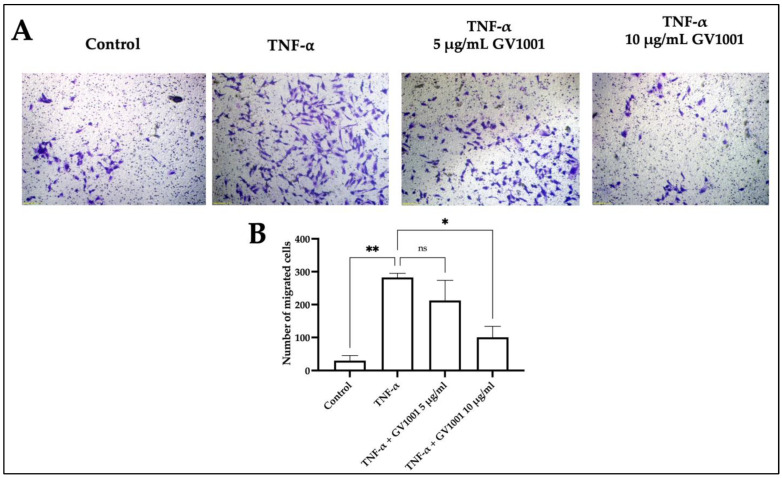
(**A**) Representative picture of migration assay of HUVEC. Migration of HUVEC exposed to 100 ng/mL TNF-α with/without 5 or 10 μg/mL of GV1001 through transwell chamber with 8-μm pore size membrane. (**B**) The number of migrated cells per field. * *p* < 0.01, ** *p* < 0.05, ns—not significantly different between two grouos (*p* > 0.05). Results represent the means ± SEM performed in four fields. The magnification of the figures were ×100.

## Data Availability

Data is available upon request to the corresponding author.

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
