# Peer review of "GV1001 Inhibits the Severity of the Ligature-Induced Periodontitis and the Vascular Lipid Deposition Associated with the Periodontitis in Mice"

_ijms, 2023, doi:10.3390/ijms241612566_

Round 1

Reviewer 1 Report

In this manuscript by Sharon Y Kim et al., authors described the effect of GV1001 on ligature-induced periodontitis and vascular lipid deposition using a mouse model. Using an in vitro cell culture model, normal and ApoE-deficient mouse model, effect of GV1001 on osteoclast formation, alveolar bone loss and proinflammatory cytokine secretion, and arterial lipid deposition were measured. Experiments are well designed to support the claim. Considering the association between periodontitis and vascular disorder, this result might be of interest to researchers in related field. I think this manuscript is suitable for publication in IJMS with minor corrections listed below.

1. line 139: "enhanced the of intensity p50" -> maybe it should be 'enhanced the intensity of p50'

2. line 295: "anti-inflammatory in mice" -> maybe 'anti-inflammatory effect in mice'?

3. line 300: NFkb -> NFκB

4. line 375: How ApoE-deficient mice were obtained should be described. C57NL/6J is normal inbred mouce line.

5. line 440: "RAW 264.76" -> please check whether it is correct

Author Response

  1. Corrected as suggested.
  2. Corrected as suggested.
  3. Corrected as suggested.
  4. Corrected as suggested

Reviewer 2 Report

Sharon Y Kim  et. al submitted a manuscript entitled”GV1001 Inhibits the Severity of the Ligature-induced Periodontitis and the Vascular Lipid Deposition associated with the Periodontitis in Mice” is interesting study to whether investigated the severity of ligature-induced periodontitis, vascular inflammation, and arterial lipid deposition in  wild-type and ApoE-deficient mice by GV1001 peptide. Interestingly, the authors showed GV1001 peptide injection reduced  the severity of  periodontitis by inhibiting gingival and systemic inflammation, alveolar bone loss. In addition, the authors also demonstrated that  GV1001 inhibited the RANKL-induced osteoclast formation and TNF-α-275 induced EndMT in endothelial cells in vitro. Finally, the authors are concluded based on the results that GV1001 peptide can use as therapeutic agent for to reduce the disease burdens of inflammatory responses caused by as periodontitis and atherosclerosis.

Author Response

Nothing to be addressed from the suggestions of the Reviewer 2.

Reviewer 3 Report

Authors have submitted quite a fascinating manuscript. The experimental design and the results seem to be exquisite and valid.

However, the some points would be considered to improve this article.

Author Response

We appreciate this reviewer's comments regarding this manuscript" "Authors have submitted quite a fascinating manuscript.  The experimental design and the results seem exquisite and valid." The figure 5 was replaced with a better jpg for better contrast, and the manuscript was thoroughly edited for improving the quality ands conclusion of the manuscript.
